# Malaria infection and anemia status in under-five children from Southern Tanzania where seasonal malaria chemoprevention is being implemented

**Richard O. Mwaiswelo**[1]*, **Bruno P. Mmbando**[2], **Frank Chacky**[3], **Fabrizio Molteni**[3], **Ally Mohamed**[3], **Samwel Lazaro**[3], **Sylvia F. Mkalla**[4], **Bushukatale Samuel**[5], **Billy Ngasala**[5]

**1** Department of Microbiology, Immunology and Parasitology, Hubert Kairuki Memorial University, Dar es Salaam, Tanzania, **2** Tanga Research Centre, National Institute for Medical Research, Tanga, Tanzania, **3** National Malaria Control Programme, Ministry of Health, Community Development, Gender, Elderly and Children, Dodoma, Tanzania, **4** Directorate of Research, Coordination, and Promotion, Tanzania Commission for Science and Technology, Dar es Salaam, Tanzania, **5** Department of Medical Parasitology and Entomology, Muhimbili University of Health and Allied Sciences, Dar es Salaam, Tanzania

* richiemwai@yahoo.com

**Data Availability Statement:** All relevant data are within the manuscript and its Supporting information files.

## Abstract

### Background

Malaria and anemia remain major public health challenges in Tanzania. Household socio-economic factors are known to influence these conditions. However, it is not clear how these factors influence malaria transmission and anemia in Masasi and Nanyumbu Districts. This study presents findings on malaria and anemia situation in under-five children and its influencing socioeconomic factors in Masasi and Nanyumbu Districts, surveyed as part of an ongoing seasonal malaria chemoprevention operational study.

### Methods

A community-based cross-sectional survey was conducted between August and September 2020. Finger-prick blood samples collected from children aged 3–59 months were used to test for malaria infection using malaria rapid diagnostic test (mRDT), thick smears for determination of asexual and sexual parasitemia, and thin smear for parasite speciation. Hemoglobin concentration was measured using a HemoCue spectrophotometer. A structured questionnaire was used to collect household socioeconomic information from parents/caregivers of screened children. The prevalence of malaria was the primary outcome. *Chi*-square tests, *t*-tests, and logistic regression models were used appropriately.

### Results

Overall mRDT-based malaria prevalence was 15.9% (373/2340), and was significantly higher in Nanyumbu (23.7% (167/705) than Masasi District (12.6% (206/1635), *p*<0.001. Location (Nanyumbu), no formal education, household number of people, household number of under-fives, not having a bed net, thatched roof, open/partially open eave, sand/soil

**Funding:** BN Global Fund Funders did not play any role in the study design, data collection and analysis, decision to publish or preparation of the manuscript.

**Competing interests:** The authors have declared that no competing interests exist.

**Abbreviations:** ACT, Artemisinin-based combination therapy; CHWs, Community Health Workers; CI, Confidence Interval; GPS, Global Positioning System; Hb, Haemoglobin; IQR, Interquartile range; ITN, Insecticide Treated Nets; mRDT, Malaria Rapid Diagnostic Test; NIMR, National Institute for Medical Research; ODK, Open Data Kit; SD, Standard deviation; SMC, Seasonal Malaria Chemoprevention; SSA, Sub-Saharan Africa; WBC, White Blood Cells.

floor, and low socioeconomic status were major risks for malaria infection. Some 53.9% (1196/2218) children had anemia, and the majority were in Nanyumbu (63.5% (458/705), $p<0.001$. Location (Nanyumbu), mRDT positive, not owning a bed net, not sleeping under bed net, open/partially open eave, thatched window, sex of the child, and age of the child were major risk factors for anemia.

## Conclusion

Prevalence of malaria and anemia was high and was strongly associated with household socioeconomic factors. Improving household socioeconomic status is expected to reduce the prevalence of the conditions in the area.

## Introduction

Malaria is an infection of major public health importance in sub-Saharan Africa (SSA) [1, 2]. It is one of the leading causes of morbidity and mortality in SSA [3]. Children aged below five years are among the groups most affected by the infection [1, 3, 4]. Anemia is one of the major complications of malaria infection [5]. Malaria infection causes hemolysis of parasitized and non-parasitized erythrocytes, and bone marrow dyserythropoiesis that compromises rapid recovery from anemia [5–7]. Consequently, in high malaria-endemic settings majority of infants and young children, and many older children and adults have anemia of varying degrees [7]. In these settings, severe life-threatening malarial anemia requiring blood transfusion in young children is among the major cause of hospital admission, especially during the rainy season [7–9]. Anemia also contributes directly or indirectly to a significant proportion of malaria-related deaths [7, 8, 10, 11]. But, effective use of malaria interventions prevents malaria morbidity including anemia, and malaria-related mortality [7]. However, the prevalence of malaria infection and anemia in Masasi and Nanyumbu districts is not well understood.

Malaria incidences, transmission rate, and vector distribution vary markedly across and within countries, or even over short distances between neighboring villages and even within a single settlement as a result of small variation in risk factors [12–14]. Understanding these risk factors may streamline the implementation and evaluation of the impact of malaria interventions. Malaria transmission is highly dependent on temperature, humidity, and rainfall [14, 15]. Besides climatic factors, malaria transmission is also influenced by other factors including socioeconomic factors, and the availability and accessibility of malaria control tools [14, 16]. Socioeconomic factors such as level of education of the head of household, occupation of the head of household, household income, type and quality of the house, and residential setting either being in rural or urban have been found to determine malaria transmission in endemic settings [13, 16]. For instance, the low level of education of the household head, low household income, poor quality of the house, late outdoor activities, and rural settings have been associated with a high burden of malaria infection [13, 16–19]. The burden of malaria is also influenced by the proximity of human settlements to vector breeding sites, and the presence of domestic animals near homesteads [14, 17, 18]. On the other hand, the availability and accessibility of the control tools significantly influence the transmission of malaria. Major tools for control of malaria in Tanzania include vector control using insecticide-treated bed nets (ITNs), indoor residual spraying, chemoprophylaxis in pregnant women, and management of clinical cases using artemisinin-based combination therapy (ACT) [20, 21]. Of the available vector control tools, ITNs are the most used tool [20–23]. The ITNs use is estimated to reduce

mortality rates by 55% in children aged below five years in sub-Saharan Africa, and incidences of malaria cases by 50% in various settings [20, 24]. To have maximum protection at the household and the entire community, high coverage with ITNs coupled with its proper use is a prerequisite [25], and one bed net per two persons per household is recommended [25, 26]. However, the socioeconomic factors influencing the transmission of malaria in Masasi and Nanyumbu districts are not well known. Likewise, the availability and accessibility of the core malaria interventions including treated bed nets are not well known. Masasi and Nanyumbu districts are planning to implement seasonal malaria chemoprevention (SMC), thus it is important to understand the prevalence of malaria and anemia in the districts, the coverage of malaria interventions such as ITNs, and the influence of socioeconomic factors in the transmission of the infection before SMC is implemented so that the additional impact of this intervention can be evaluated. This article, therefore, presents findings to enlighten the baseline malaria and anemia situation and the influencing socioeconomic factors in Masasi and Nanyumbu districts before the implementation of SCM.

## Materials and methods

### Study area

The study was carried out in Nanyumbu and Masasi Districts, Mtwara Region. Nanyumbu has a total area of 5,204 square km and is divided into 14 wards and 89 villages, whereas Masasi has a total area of 3,829.9 square km and is divided into 22 wards and 159 villages. Nanyumbu District has a total population of 166,277, and Masasi has 269,590. Of the population, 115,735 and 152,609 in Nanyumbu and Masasi Districts, respectively, live in rural areas. The districts had a projected population of 82,740 children aged 3–59 months in 2019, 31,564 in Nanyumbu, and 51,176 in Masasi District. By 2018, there were 44,319 and 73,998 households in Nanyumbu and Masasi District, respectively. Nanyumbu has 19 health facilities and Masasi has 40. The major economic activities in Masasi and Nanyumbu Districts include subsistence farming, trade, fishing, and small-scale mining.

Nanyumbu and Masasi Districts have annual rainfall averaging 939 mm and a temperature of 25.4˚C. The rainy season in Nanyumbu and Masasi Districts is between January and April and accounts for > 60% of the average annual rainfall. Both districts have high seasonal malaria transmissions where > 60% of cases occur between March and July. *P. falciparum* is the predominant malaria parasite, and *Anopheles arabiensis* the major vector. Insecticide-treated bed-nets and diagnosis and treatment with antimalarial drugs are the major malaria control measures in the area.

### Study design

This was a community-based cross-sectional malariometric and socioeconomic survey with wards used as sampling units. The wards were selected using a Research Randomizer software version 4 (Wesleyan University, Connecticut, USA) [27]. Health facilities nearest to the centroid of the selected wards were identified (satellite facilities) and the catchment population was estimated. A study village(s) within the health facility catchment area was identified (satellite villages) and the demographic survey was carried out to determine the population structure, and the number of possible study participants as well as socioeconomic characteristics of the households. A malariometric survey was conducted in all the selected villages to establish baseline malariometric indices including the type of malaria parasite, the prevalence of malaria infection, hemoglobin levels, and assessment of nutritional status.

## Study population

Both febrile and afebrile children of both sexes were involved in the malariometric survey, whereas heads of the households were involved in the household survey. The study inclusion criteria for the malariometric survey were being a child aged 3–59 months, living within the catchment area, and the willingness of the parents/caregivers to participate in the study. The exclusion criteria were the presence of severe illness, and history of intake of antimalarial drug within the previous 30 days or being under cotrimoxazole prophylaxis.

Inclusion criteria for the household survey included living within the catchment area for at least 5 years, willingness to participate in the study, and having a child who has participated in the malariometric survey. The head of the household was defined as a person who is perceived by household members to be the primary decision-maker in the family, while a household was defined as individuals living together and taking meals from a common cooking facility [28]. In the absence of the head of the household, a responsible person above 18 years who had been appointed by the family was interviewed.

## Procedures for data collection

Research assistants including clinicians, laboratory technicians, and community health workers were trained on the study objectives and procedures before data collection. Social mobilization of the participating communities was organized through local meetings and announcements in the houses of worship before the start of the study to explain the purpose of the study, its benefit to the community, and the importance of community members' participation.

**Malariometric survey.** The survey was carried out at the satellite health facilities whereby all the febrile and afebrile children of the required age were brought for clinical and laboratory assessments. The survey was conducted for three days at each of the selected wards.

The clinical assessment involved taking a history of clinical symptoms, history of antimalarial drugs consumption within the past 30 days, use of cotrimoxazole as chemoprophylaxis, clinical examination including measurement of axillary temperature, mid-upper-arm circumference (MUAC), and anthropometric (height and weight) measurements. The anthropometric measurements were recorded for each child by the same investigator using standard techniques [29, 30]. The height of the children was recorded to the nearest 0.1 cm using an anthropometric height rod. For children below 24 months of age, length was measured using an infant meter. The weight measurements were recorded to the nearest 100 g using SECA electronic weighing scale. Repeated measurements were made for 20 children to periodically correct for the intraobserver error. The z-scores for three indices, i.e., weight-for-age, height-for-age, and weight-for-height were then calculated.

Laboratory assessment involved collection of finger-prick blood samples that were used to assess the presence of malaria infection using malaria rapid diagnostic test (mRDT), preparing the thick film for microscopy to assess the density of asexual parasitemia, and gametocytaemia, and thin-film for asexual parasite species determination. The blood samples were also used to measure hemoglobin concentration.

Thin films were fixed using absolute methanol. Both thin and thick blood films were air-dried, stained using 3% Giemsa for 1 hour, and examined for malaria parasites at 100 high power fields under immersion oil. Parasite density was determined by counting the number of parasites present per 200 white blood cells (WBC) on a thick smear, and the obtained number was multiplied by 40 assuming a WBC count of 8,000 per milliliter of blood. A blood slide was considered negative if no parasite was seen after examining 100 fields. Slides were read independently by 2 laboratory technicians, who were unaware of the ward the slides came from

and the mRDT results. In case of a discrepancy (positive versus negative or a difference in parasite density greater than 30%), a third reading was requested and the average parasite density of three readings was used in case of difference in parasite density of greater than 30%, whereas the average of the two positive readings was used for the case of positive versus negative results. Microscopists at the National Institute for Medical Research (NIMR), Tanzania read 10% of the slides for quality control.

Hemoglobin concentration was measured using a portable spectrophotometer, HemoCue Hb 301+ (HemoCue AB, Ängelholm Sweden), with a precision of +/- 0.3 g/dL. The HemoCue was calibrated every morning using a control cuvette at 16.0+/-0.3 g/dL according to manufacturer's instruction [31]. Anaemia was classified as haemoglobin level < 11g/dL (mild), < 7g/dL (moderate) and < 5g/dL (severe).

**Socioeconomic survey.**   The survey was carried out in catchment villages of the selected wards. The Community Health Workers (CHWs) conducted the household survey using a structured questionnaire with both close and open-ended questions. The CHWs administered the questionnaire. The questionnaire inquired information on demographic characteristics of the household, socioeconomic status including house type and household assets, awareness on malaria infection and its control measures, and ownership, usage, and perception of the household on the available major malaria control tools in the districts particularly insecticide-treated bed nets (ITN). The use of ITN was assessed by asking if a child slept under ITN the previous night, and the CHWs confirmed the presence of the bed net.

## Study outcomes

The primary outcome was the prevalence of malaria infection defined as the presence of *P. falciparum* asexual parasitemia at any density. Secondary outcomes include (i) mild, moderate, or severe anemia defined as a hemoglobin concentration of 11g/dL, 8g/dL, and 5g/dL, respectively; (ii) clinical presentations that predict malaria infection (iii) socioeconomic factors influencing malaria transmission (iv) socioeconomic factors influencing anemia.

## Ethical consideration

The study was conducted following the declaration of Helsinki, good clinical practices, and regulations in Tanzania [32]. The ethics committee of the Muhimbili University of Health and Allied Sciences, Tanzania, approved the study.

Members of the study team held meetings with community, administrative, and religious leaders to explain the aims and activities of the study and sought community approval. Project CHWs then visited the households to explain the aim of the study, provided information sheets, and sought signed consent from parents or guardians of the children who were involved in the malariometric survey. The parents/guardians also provided consent to participate in the socioeconomic survey.

## Statistical analysis

This was a baseline survey for seasonal malaria chemoprevention (SMC) study, thus sample size calculation was based on the estimations made for SMC, and is presented elsewhere. Briefly, there is variation in malaria incidence between the two districts, with three years (2016–2018) average of 309 and 222.5 cases per 1000 in Nanyumbu and Masasi District, respectively. Malaria control in both districts relies mainly on long-lasting insecticides nets and prompt diagnosis and treatment with ACTs for uncomplicated *falciparum* malaria. Thus a total of 20 wards were selected for the study, and in each ward 106 children were to be assessed with a power of 80% and an alpha (type one error) of 0.05. An attrition rate of 20% was

considered resulting in a sample of 128 children per evaluable ward. Similar to the number of children expected to participate in the malariometric survey, 128 households had to be assessed for socioeconomic survey assuming each household contributed one child.

Data were collected electronically in open data kit (ODK) software using tablets computers. Spatial data (point coordinates for households and boundaries of villages and other features from the study areas) were collected during the household visits using the Global Positioning System (GPS) inbuilt in mobile phones. Quality control and assurance of the data were maintained at all stages of data collection to archiving. Back-ups of data were made daily onto external hard disks and stored in a secured place separately from the building hosting data management section. Source documents (in case of paper-based) were achieved and sprayed regularly to keep them away from destruction by insects for the duration specified in the protocol. Data were cleaned and analyzed using STATA version 13.1 statistical package, while analysis was done using STATA and R (https://cran.r-project.org/). The z-scores for three indices, i.e., weight-for-age, height-for-age, and weight-for-height were calculated using *the zanthro* function in STATA, and the WHO standards were used [29, 30]. Categorical variables were presented in proportions and compared using $\chi^2$-tests while continuous variables were summarised as means with standard deviations and compared using t-test. Mean values and 95% confidence intervals (95%CI) for different variables were compared between mRDT positive and negative individuals using a t-test. A $\chi2$ test was also used to assess the trend of continuous and categorical variables by age groups. Logistic regression analysis was used to assess the association of mRDTs positivity with different parameters. A $p$-value $< 0.05$ was considered statistically significant.

## Results

### Baseline characteristics and distribution of malaria infection by age and district

A total of 2340 children, 1635 (69.9%) from Masasi and 705 (30.1%) from Nanyumbu Districts were screened for malaria parasites using both mRDT and microscopy. The baseline characteristics of the screened children are presented in Table 1. Of the screened children, 373/2340 (15.9%) were malaria positive by mRDT and 212/2330 (9.1%) were positive by microscopy. The prevalence of malaria was significantly higher in Nanyumbu (23.7% (167/705 than in Masasi district (12.6% (206/1635), ($p<0.001$).

**Table 1. Baseline characteristics of the participants.**

| Variable | Overall | Masasi | Nanyumbu | Test |
|---|---|---|---|---|
| Age, median (IQR), years | 2.25 (1.12–3.43) | 2.23 (1.18–3.45) | 2.24 (1.27–3.40) | $p = 0.855$* |
| Sex, female, n (%) | 1215 (51.92) | 865 (52.91) | 350 (49.65) | $p = 0.148$ |
| Weight (kg), mean (SD) | 11.45 (2.99) | 11.52 (3.02) | 11.29 (2.91) | $p = 0.086^{\delta}$ |
| Height (cm), mean (SD) | 81.72 (13.18) | 81.99 (12.96) | 81.13 (13.64) | $p = 0.144^{\delta}$ |
| mRDT positive, n (%) | 373 (15.94) | 206 (12.60) | 167 (23.69) | $p<0.001$ |
| Blood slide positive, n (%) | 212 (9.10) | 106 (6.52) | 106 (15.08) | $p<0.001$ |
| Mean parasite density/µL, (95%CI) | 193 (137–272) | 458 (268–785) | 81 (57–117) | $p<0.001$ |
| Mean Hb concentration, g/dL, (SD) | 10.94 (1.42) | 11.1 (1.43) | 10.6 (1.34) | $p<0.001^{\delta}$ |

*Kruskal-Wallis equality of population rank test;
$\delta$two sample t-test.

**Table 2. Distribution of malaria infection by age group and district.**

| Age group (months) | mRDT malaria positivity | | | Test, $p$-value |
|---|---|---|---|---|
| | Overall (%) | Masasi (%) | Nanyumbu (%) | |
| 3–11 | 49/474 (10.3) | 26/337 (7.7) | 23/137 (16.8) | $x^2 = 8.65, p = 0.003$ |
| 12–23 | 63/511 (12.33) | 28/333 (8.41) | 35/178 (19.66) | $x^2 = 13.59, p <0.001$ |
| 24–35 | 89/482 (18.46) | 52/336 (15.48) | 37/146 (25.34) | $x^2 = 6.58, p = 0.010$ |
| 36–47 | 86/425 (20.24) | 48/294 (16.33) | 38/131 (29.01) | $x^2 = 9.03, p = 0.003$ |
| 48–59 | 82/352 (23.30) | 48/241 (19.92) | 34/111 (30.63) | $x^2 = 4.88, p = 0.027$ |
| **Trend test** | $x^2 = 27.1, p<0.001$ | $x^2 = 16.3, p<0.001$ | $x^2 = 11.2, p<0.001$ | |

The distribution of malaria infection by age groups and district is presented in Table 2. The prevalence of malaria was highest in the age group of 48–59 months both in the overall study population and by districts. Trend analysis showed that the prevalence of malaria was statistically significantly increased with age in the overall study population and by the district.

## Relationship between clinical presentation and malaria infection status

The relationship between clinical presentation and mRDT positivity was assessed in both symptomatic and asymptomatic children. The likelihood of mRDT tests to give positive results was statistically significantly higher in children with symptoms than in those without symptoms, Table 3.

## Relationship between socioeconomic factors and malaria infection

Socioeconomic factors and their relationship with malaria infection are presented in Table 4. In a univariate analysis factors including district (Nanyumbu), education level of the head of household, having five or more occupants in the household, having more than one under-five

**Table 3. Relationship between clinical presentation and mRDT positivity.**

| Clinical presentation | | mRDT positive (%) | Test, $p$-value |
|---|---|---|---|
| Fever | Yes | 148/214 (69.2) | $x^2 = 482, p<0.001$ |
| | No | 222/2048 (10.8) | |
| History of fever in past 24 hours | Yes | 245/284 (86.3) | $x^2 = 1200, p<0.001$ |
| | No | 125/1934 (6.5) | |
| Use of antimalarial drugs within past 14 days | Yes | 201/478 (42.0) | $x^2 = 282, p<0.001$ |
| | No | 169/1740 (9.7) | |
| Headache | Yes | 101/107 (94.4) | $x^2 = 488, p<0.001$ |
| | No | 269/2111 (12.7) | |
| Cough | Yes | 119/355 (33.5) | $x^2 = 86.2, p<0.001$ |
| | No | 251/1863 (13.5) | |
| Diarrhea | Yes | 23/41 (56.1) | $x^2 = 47.0, p<0.001$ |
| | No | 347/2177 (15.9) | |
| Vomiting | Yes | 58/64 (90.6) | $x^2 = 259, p<0.001$ |
| | No | 312/2154 (14.5) | |
| Abdominal pain | Yes | 88/92 (95.7) | $x^2 = 430, p<0.001$ |
| | No | 282/2126 (13.3) | |

**Table 4. Relationship between household socioeconomic factors and malaria infection.**

| Variable | | | Univariate | Multivariate (n = 1203) |
|---|---|---|---|---|
| | | Prevalence (%) | OR (95%CI); *p*-value | aOR (95%CI) *p*-value |
| District | Masasi | 206/1652 (12.5) | 1 | 1 |
| | Nanyumbu | 168/716 (23.5) | 2.15 (1.71–2.70); <0.001 | 1.9 (1.34–2.68); < .001 |
| Sex of head | Male | 197/1,237 (15.9) | 1 | |
| | Female | 66/429 (15.4) | 0.96 (0.71–1.30); 0.791 | |
| Marital status | Single | 72/438 (16.4) | 1 | |
| | Married | 176/1,137(15.5) | 0.93 (0.69–1.26); 0.640 | |
| | Separates | 15/90 (16.7) | 1.02 (0.55–1.87); 0.958 | |
| Education level | None | 36/142(25.3) | 1 | 1 |
| | Any primary | 217/1,374(15.8) | 0.55 (0.37–0.83); 0.004 | 0.53 (0.31–0.90); 0.018 |
| | Any Secondary | 10/150 (6.7) | 0.21 (0.10–0.46); < .001 | 0.33 (0.13–0.81); 0.016 |
| Occupation | Farming | 245/1,548(15.8) | | |
| | Petty trader | 18/105 (17.1) | 1.10 (0.65–1.86); 0.721 | |
| No. of people | 2–5 people | 197/1,322 (14.9) | 1 | 1 |
| | >5 people | 66/339 (19.5) | 1.38 (1.01–1.88); 0.040 | 1.52 (1.02–2.25); 0.039 |
| No. children <5 years | One | 190/1,322 (14.3) | 1 | 1 |
| | More than One | 70/284 (24.6) | 1.96 (1.43–2.68); < .001 | 1.39 (0.92–2.10); 0.120 |
| Ownership of bed nets | Yes | 185/1,412 (13.1) | 1 | 1 |
| | No | 78/254(30.7) | 2.95 (2.16–4.03); < .001 | 2.53 (1.60–4.0); < .001 |
| Who use bed nets | Child plus others | 179/1,394 (12.8) | 1 | |
| | Parents only | 6/18 (33.3) | 3.39 (1.25–9.18); 0.010 | |
| Roof type | Thatch | 140/595 (23.5) | 1 | |
| | Iron sheet/tiles | 123/1,071 (11.5) | 0.42 (0.32–0.52); < .001 | |
| Eave type | Open | 184/1,226 (15.0) | 1 | |
| | Partially open | 30/86 (34.9) | 0.30 (0.17–0.52); < .001 | |
| | Closed | 49/354 (13.8) | 0.32 (0.20–0.53); < .001 | |
| Window type | No window | 72/468(15.4) | 1 | |
| | Wire mesh | 60/388 (15.5) | 1.01 (0.70–1.46); 0.974 | |
| | Wood/iron sheet | 96/638(15.1) | 0.97 (0.70–1.36); 0.877 | |
| | Thatch | 35/172(20.3) | 1.41 (0.90–2.20); 0.136 | |
| Floor-type | Soil/sand | 192/1,069 (18.0) | 1 | |
| | Cement | 71/597 (11.9) | 0.62 (0.46–0.83); 0.001 | |
| Presence of electricity | Yes | 14/127 (10.2) | | |
| | No | 249/1,539 (16.2) | 1.56(0.88–2.76); 0.126 | |
| Social economic status | Low | 83/409 (20.3) | 1 | 1 |
| | Medium | 50/409 (12.2) | 0.55 (0.37–0.80); 0.002 | 0.54 (0.36–0.83); 0.005 |
| | Upper | 50/421 (11.9) | 0.53 (0.36–0.78); 0.001 | 0.41 (0.25–0.66); < .001 |

OR = odds ratio; aOR = adjusted odds ratio; 95%CI = 95% confidence interval.

in the household, ownership of a bed net, only parents using bed net, roof type, floor type, and household socioeconomic level were the factors statistically significantly associated with malaria infection. Multivariate model analysis indicated that district (Nanyumbu), more than 5 people living in the household were positively associated with a high risk of malaria infection while having primary or secondary education, and medium and upper economic status was positively associated with low risk of infection.

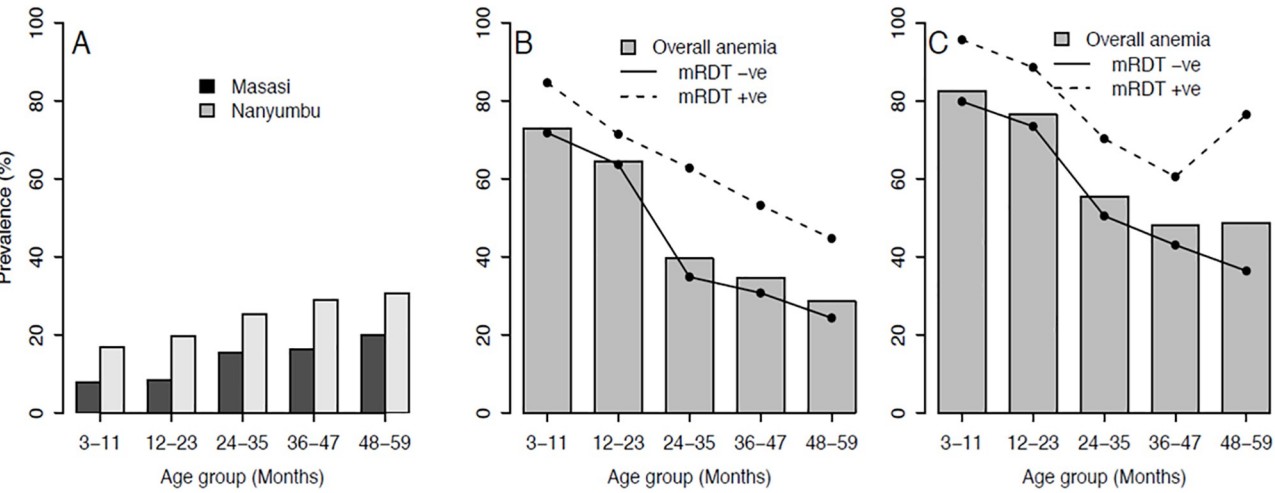

**Fig 1. Prevalence of mRDT positives (A) and anemia (B) by district and by age group.** The open bars are Masasi, filled bars in Nanyumbu District while line segments present 95%CIs.

### Prevalence of anemia and its distribution in relation to malaria infection status

A total of 53.9% (1196/2218) of screened children had anemia, of which 49.4% (748/1513) were in Masasi and 63.5% (458/705) in Nanyumbu District, ($x^2$ = 38.5, $p < 0.001$). Of the children with anemia, 95.9% and 4.1%, had mild and moderate anemia, respectively. Fig 1 shows the prevalence of malaria by age groups in the two districts (Panel A) and that of anemia by age groups in relation to malaria infection status in Masasi (Panel B) and Nanyumbu District (Panel C). The prevalence of anemia was significantly higher in children with mRDT positive results than in those with negative results both in Masasi (61.1% (124/203 *vs* 48.0% (636/1326) $x^2$ = 12.1, $p < 0.001$) and in Nanyumbu District (76.8% (129/168) *vs* 59.9% (328/548), $x^2$ = 16.0, $p < 0.001$). The prevalence of anemia was significantly decreasing with an increase in age.

### Relationship between socioeconomic factors and anemia

The relationship between household socioeconomic factors and anemia in the study population is presented in Table 5. Using the univariate analysis, the participant's district, mRDT positivity, ownership of bed nets, eave type, sex of the child, age of the child, and thatched window were the factors associated with anemia in children. In the multivariate model only District, mRDT positivity, sex of the child, age of the child, and floor type were the factors associated with anemia.

## Discussion

Malaria is still an infection of major public health importance in Tanzania, especially among under-five children [20, 22]. This study provides important information on the prevalence of malaria infection and anemia among under-fives and the associated household socioeconomic factors in Masasi and Nanyumbu Districts, the area where malaria transmission is highly seasonal, and seasonal malaria chemoprevention operational research is ongoing. The overall prevalence of malaria in this area was high at 15.94%, and it was two folds higher than the national average of 7.3% [20]. But malaria prevalence in this study area was lower than that in Kakonko (30.8%) and Nanyamba (19.1%) Districts, Tanzania [20]. Of the socioeconomic

**Table 5. Household socioeconomic factors associated with anemia.**

| Variable | | Univariate | | Multivariate |
|---|---|---|---|---|
| | | Prevalence (%) | OR (95%CI); *p*-value | aOR (95%CI); *p*-value |
| District | Masasi | 748/1513 (49.4) | 1 | 1 |
| | Nanyumbu | 448/705 (63.5) | 1.78 (1.48–2.15); <0.001 | 1.97 (1.55–2.50); <0.001 |
| mRDT results | Negative | 944/1848 (51.1) | 1 | 1 |
| | Positive | 252/370 (68.1) | 2.05 (1.61–2.60); <0.001 | 2.34 (1.72–3.19); <0.001 |
| Marital status | Single | 240/421 (57.0) | 1 | |
| | Married | 580/1056 (54.9) | 0.92 (0.73–1.15); 0.467 | |
| | Separated | 42/89 (47.2) | 0.67 (0.43–1.07); 0.09 | |
| No. of people | 2–5 | 684/1251 (54.7) | 1 | |
| | >5 | 175/312 (56.1) | 1.06 (0.82–1.36); 0.654 | |
| Ownership of bed net | Yes | 715/1327 (53.9) | 1 | |
| | No | 147/240 (61.2) | 1.35 (1.02–1.79; 0.035 | |
| Who use a bed net | None | 147/240 (61.2) | 1 | |
| | All | 705/1310 (53.8) | 0.74 (0.56–0.98); 0.033 | |
| | Parents only | 10/17 (58.8) | 0.904 (0.33–2.46); 0.843 | |
| Roof type | Thatch | 329/575 (57.2) | 1 | |
| | Iron sheet | 533/992 (53.7) | 0.87 (0.71–1.07); 0.181 | |
| Eave type | Open | 61/85 (71.8) | 1 | |
| | Partially open | 174/302 (57.6) | 0.53 (0.32–0.91); 0.018 | |
| | Closed | 627/1180 (53.1) | 0.45 (0.27–0.73); <0.001 | |
| Window type | No window | 243/424 (57.3) | 1 | |
| | Wire mesh/gauze | 192/364 (52.7) | 0.83 (0.63–1.10); 0.199 | |
| | Wood/iron sheet | 345/608 (56.7) | 0.98 (0.76–1.26); 0.856 | |
| | Thatch | 82/171 (47.9) | 0.69 (0.48–0.98); 0.038 | |
| Floor type | Soil/sand | 574/1012 (56.7) | 1 | 1 |
| | Cement | 288/555 (51.9) | 0.82 (0.67–1.01); 0.066 | 0.77 (0.61–0.98);0.033 |
| Sex of child | Male | 629/1072 (58.7) | 1 | 1 |
| | Female | 567/1146 (49.5) | 0.69 (0.58–0.82); <0.001 | 0.63 (0.51–0.79); 0.001 |
| Age of child in years | 3–11 | 341/450 (75.8) | 1 | 1 |
| | 12–23 | 333/484 (68.8) | 0.70 (0.53–0.94); 0.017 | 0.64 (0.45–0.91); 0.014 |
| | 24–35 | 200/447 (44.7) | 0.26 (0.19–0.35); <0.001 | 0.21 (0.15–0.30); <0.001 |
| | 36–47 | 158/406 (38.9) | 0.20 (0.15–0.28); <0.001 | 0.18 (0.13–0.27); <0.001 |
| | 48–59 | 118/335 (35.2) | 0.17 (0.12–0.24); <0.001 | 0.13 (0.09–0.19); <0.001 |
| Economic status | Low | 223/395 (56.5) | 1 | |
| | Medium | 214/383 (55.9) | 0.98 (0.74–1.30); 0.870 | |
| | Upper | 214/386 (55.4) | 0.96 (0.72–1.127); 0.775 | |

OR = odds ratio; aOR = adjusted odds ratio; CI = confidence interval

factors assessed in relation to malaria infection at the household level, a location particularly living in Nanyumbu District, having no education, large family size (>5 people), household having more than one under-five children, only parents sleeping under the bed net, a household not owning a bed net, thatched roof, sand/soil floor, open/partial open eave, and low socioeconomic status of the household were significantly associated with an increased risk of malaria infection. Similar factors have been associated with increased risk of malaria infection in other parts of Tanzania [13, 33–36], and in other countries including Botswana [18], Cameroon [37], Ethiopia [38, 39], Equatorial Guinea [40], India [41], Kenya [42], and Rwanda

[43, 44]. The level of education affects the overall knowledge and ability of an individual to make a decision, and also affects occupation and income. In this study, probably having no education led to poor decision making towards malaria control measures, and or poor income thus could not afford the control tools such as bed net, hence increasing the risk of infection. Large family size and low socioeconomic status could probably lead to the family failing to afford the adequate number of bed nets and other commodities needed to fight malaria. Besides ITNs providing a physical barrier, the insecticide they contain has repelling and knockdown effects against *Anopheles* mosquitoes, thus improving the control of the infection [25]. On the other hand, a large number of people in the house/room leads to increased production of carbondioxide and other volatile gases that attract mosquitoes thus increasing the number of mosquitoes in the house, and also many people are exposed to infectious bites per night [45–47]. Open/partial open eaves allows for easy entry of mosquitoes into the house, whereas thatched roof and sand/mud floor provide conducive microhabitat for mosquito resting [47], thus increasing the risks of an infectious mosquito bite to the household occupants. While most socioeconomic factors can be associated with malaria infection and anemia in the cause relation path; malaria and anaemia too may have contributed to poor socioeconomic status observed in the study area, contrary to our discussion which relied on the first scenario. This complex relationship requires careful consideration in the analysis and interpretation of the research findings.

Nonetheless, in this study occupation, window type, and electricity were not associated with malaria infection. Contrarily, in other studies [41, 42, 47], these factors were positively associated with malaria infection. Occupation and electricity had no significant influence on malaria infection probably because nearly all of the household heads had the same occupation, they were peasants, and likewise, most of the households (92.4%) had no electricity. On the other hand, malaria prevalence in Nanyumbu District was nearly two folds higher than that of Masasi District. It is, however, not clear why Nanyumbu had a much higher malaria prevalence than Masasi. All the assessed socioeconomic factors were not statistically significantly different between the two districts. However, the differences in malaria prevalence may probably be attributed to differences in transmission dynamics between the two districts including mosquito abundance, the factor which was not assessed in this study.

The prevalence of malaria was increasing with an increase in age. Similar findings have been reported in other studies [16, 40, 43, 48, 49]. The low infection rate among neonates and young infants might be attributed to the passive immunity infants acquire from their mothers [50, 51], and the fetal hemoglobin [50]. Passive immunity protects the infants against many infections including malaria, but as they grow this immunity tends to wane off, and thus children become prone to the infection [50, 51]. Fetal hemoglobin is known to poorly support *Plasmodium* parasite survival [50]. Furthermore, probably at a younger age many infants and younger children sleep under the bed nets, and normally under the protection of their parents who can control their sleeping behavior. But as they grow they stop sleeping with their parents, and therefore, even though they might be sleeping under the bed net, it becomes difficult to control their sleeping behavior and thus expose themselves to the mosquito bite.

In this study, clinical presentations were strongly associated with mRDT positivity. Of the clinical presentations, abdominal pain (95.7%), headache (94.4%), vomiting (90.8%), history of fever in the past 24 hours (86.3%), and fever (69.3%) were the strongest predictors of mRDT positivity. Other studies have also reported fever [52, 53], history of fever [53], headache [53], and cough [35] to be the predictors of malaria infection. Interestingly, being a community-based study it was expected that there would be no strong association between clinical presentations and mRDT positivity. However, in malaria-endemic settings, asymptomatic malaria infection is common particularly in older children and adults who normally have developed

partial immune against the infection, the immunity which normally develops after years of exposure to the infection. But since the children in this study were under the age of five years, probably they had not developed enough immunity to partially protect and lender them asymptomatic upon the infectious bite.

Anemia is one of the major complications of malaria infection, contributing directly or indirectly to hospitalization and deaths in young children [8–10]. More than half of the screened children in this study had anemia, and the majority of them (95.9%) had mild anemia. Socioeconomic factors such as being a resident of Nanyumbu District, mRDT positive results, ownership of bed nets, not sleeping under bed net, open/partially open eave, no windows, thatched window, sand/soil floor, male sex, and age of the child were significantly associated with anemia. The same factors are associated with anemia in other studies [16, 42, 48, 49]. However, whereas in other studies having no windows was not associated with malaria infection, in this study it was associated with increased risk of infection. Not having or not sleeping under a bed net increases exposure to mosquitoes, whereas open/partial open eaves allow easy entry of mosquitoes into the house. Not having windows, thatched windows, and sand/soil floor provides a good environment for mosquitoes. These socioeconomic factors either individually or collectively increase exposure to malaria infection that in turn leads to anemia. Contrary to the findings in other studies [37, 44], in this study socioeconomic status and number of people in the household were not associated with anemia. Conversely, the prevalence of anemia was much higher in Nanyumbu (63.9%) than in Masasi District (49.6%). Malaria infection may probably be the reason for the higher prevalence of anemia in Nanyumbu as the infection was more prevalent in Nanyumbu District than Masasi District. On the other hand, age and sex also influence the occurrence of anemia especially in girls when they reach puberty and start to menstruate. However, in this study all the involved children were under the age of five years, hence being a girl could not influence the occurrence of anemia. But male sex was associated with anemia probably because of the boys' behaviors such as playing outside till late evening that may expose them to malaria infection and in turn anemia. Importantly, anemia lowers the body's immunity and in turn, may lead to increased exposure to malaria infection [54, 55].

The prevalence of anemia was significantly higher in children with mRDT positive results than in those with negative results. Studies in Ghana [48, 49], Kenya [16], and Rwanda [43, 44] have presented similar findings. On the other hand, whereas the prevalence of malaria was increasing with an increase in age, the prevalence of anemia was decreasing with an increase in age. Similar findings have been reported in other studies [49, 56]. The decrease of anemia with an increase in age probably indicated that at a younger age, neonates and young infants are not breastfed adequately, and thus nutrients obtained in the milk are not adequate. But as these infants grow and start to feed on solid food they get more nutrients that improve their hemoglobin level, hence reducing the prevalence of anemia. Interestingly, more than half of the mRDT negative children had anemia. Of note, most parts of Tanzania including Nanyumbu and Masasi are endemic to intestinal soil-transmitted helminths which are also among the major causes of anemia in children living in tropical countries [57–60]. Likewise, malnutrition and red blood cell disorders including sickle cell anemia are other factors that may have contributed to the observed high prevalence of anemia in non-malaria-infected children [56].

Despite the strength of this study, its limitations include: the children were not screened for soil-transmitted helminths which are also a significant cause of anemia in tropical countries. Red blood cells disorders were also not assessed and are known to cause anemia. Nutritional assessment was performed, however will be presented elsewhere. Mosquito abundance was not assessed, and this could have probably explained the significant differences in malaria

prevalence between Masasi and Nanyumbu districts. Nonetheless, the authors believe that these limitations have not affected the validity of the study findings.

## Conclusion

The prevalence of malaria and anemia was high among under-five children in Masasi and Nanyumbu districts and was strongly associated with household socioeconomic factors. Improving the household socioeconomic status is expected to reduce the prevalence of malaria and anemia in the study area.

## Supporting information

**S1 Data.**
(XLSX)

## Acknowledgments

The authors extend their appreciation to the children and parents/guardians for participating in the study. We would also like to extend our appreciation to the local authorities for their support in all steps of this study.

## Author Contributions

**Conceptualization:** Richard O. Mwaiswelo, Bruno P. Mmbando, Frank Chacky, Fabrizio Molteni, Ally Mohamed, Samwel Lazaro, Sylvia F. Mkalla, Billy Ngasala.

**Data curation:** Richard O. Mwaiswelo, Bruno P. Mmbando, Billy Ngasala.

**Formal analysis:** Richard O. Mwaiswelo, Bruno P. Mmbando.

**Funding acquisition:** Frank Chacky, Fabrizio Molteni, Ally Mohamed, Samwel Lazaro, Sylvia F. Mkalla, Billy Ngasala.

**Investigation:** Richard O. Mwaiswelo, Bruno P. Mmbando, Bushukatale Samuel, Billy Ngasala.

**Methodology:** Richard O. Mwaiswelo, Bruno P. Mmbando, Billy Ngasala.

**Project administration:** Richard O. Mwaiswelo, Bruno P. Mmbando, Bushukatale Samuel, Billy Ngasala.

**Resources:** Richard O. Mwaiswelo, Bruno P. Mmbando, Bushukatale Samuel, Billy Ngasala.

**Supervision:** Richard O. Mwaiswelo, Bruno P. Mmbando, Billy Ngasala.

**Validation:** Richard O. Mwaiswelo, Bruno P. Mmbando, Billy Ngasala.

**Visualization:** Billy Ngasala.

**Writing – original draft:** Richard O. Mwaiswelo.

**Writing – review & editing:** Richard O. Mwaiswelo, Bruno P. Mmbando, Frank Chacky, Fabrizio Molteni, Ally Mohamed, Samwel Lazaro, Sylvia F. Mkalla, Bushukatale Samuel, Billy Ngasala.

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
