## [Decision Letter · Decision Letter 0]

14 Sep 2021

PONE-D-21-21864Malaria infection and anemia status in underfive children in an area initiating seasonal malaria chemoprevention in Southern Tanzania.PLOS ONE

Dear Dr. Mwaiswelo,

After careful consideration, we feel that your manuscript will likely be suitable for publication if the authors revise it to address specific points raised by the reviewers. According to the reviewers, there are some specific areas where further improvements would be of substantial benefit to the readers.   For your guidance, a copy of the reviewers' comments was included below. 

We look forward to receiving your revised manuscript.

Kind regards,

Luzia Helena Carvalho, Ph.D.

Academic Editor

PLOS ONE

Journal Requirements:

2. Thank you for stating the following in the Acknowledgments/ Funding Section of your manuscript: 

The study was funded by Global Fund.

BN

Global Fund

Funders did not play any role in the study design, data collection and analysis, decision to publish or preparation of the manuscript.

No authors have competing interests

Reviewers' comments:

Reviewer's Responses to Questions

**Comments to the Author**

1. Is the manuscript technically sound, and do the data support the conclusions?

Reviewer #1: Yes

Reviewer #2: Yes

2. Has the statistical analysis been performed appropriately and rigorously? 

Reviewer #1: I Don't Know

Reviewer #2: Yes

3. Have the authors made all data underlying the findings in their manuscript fully available?

Reviewer #1: Yes

Reviewer #2: Yes

4. Is the manuscript presented in an intelligible fashion and written in standard English?

Reviewer #1: Yes

Reviewer #2: No

5. Review Comments to the Author

Reviewer #1: My compliments to the authors, this is a well written manuscript about a properly designed and explained study. However, I have a few questions/remarks though:

Title

- The title was not fluent. So I wanted to suggest “Malaria infection and anemia status in underfive South Tanzanian children where seasonal malaria chemoprevention is being implemented” instead of “Malaria infection and anemia status in underfive children in an area initiating seasonal malaria chemoprevention in Southern Tanzania”.

-Under introduction

Justification and rationale for the study were not clearly stated. However, the introduction was more of information from literature. Please kindly review your introduction by elaborating on your justifications and rationales.

- Was the Ward randomization performed with a computer? Please elaborate.

- Under study population

- I will suggest you rephrase it (P6, line 127) to mean willingness of the parents or caregivers to consent since these infants and your children are ethnically ineligible to consent for the study themselves.

- P6, line 128, please add the duration of stay in the study area.

- Under malariometric study

- Please cite (P7, lines 148-149).

- Please specify software program used and name of manufacturer (P7, line 155).

- Please specify the number of readings used in the calculation for average parasite density (P8, lines 170- 171).

- Please cite (P8, lines 174-175).

- Under ethical consideration

- Please cite (P8 line 194-195).

- Under relationship between clinical presentation and malaria infection status

- The sentence is not clear (P12, lines 259-263).

- Please clearly state whether it is positively or negatively associated (P13, line 266).

- See Attachment for others minor errors.

Reviewer #2: Congratulations on the manuscript and the intense work carried out in the field.

- The study is an analysis of descriptive data in two endemic regions for malaria and anemia in Africa, in which socioeconomic factors were evaluated, mainly those related to housing, and the condition of clinical and laboratory malaria and also the prevalence of anemia. The method, results and analysis are well presented. The discussion and conclusion well represent the results, which despite not adding new concepts, are important for your region, confirming and describing the prevalence of social variables related to malaria and anemia. A better discussion about the presence of interfering or confounding variables in the results or even the complex causal relationship between socioeconomic data and malaria and anemia. When, socioeconomic conditions can be the cause of malaria and anemia, but also malaria and anemia are causes of low socioeconomic condition.

- The manuscript needs language corrections, mainly in the introduction and methodology.

6. PLOS authors have the option to publish the peer review history of their article (what does this mean?). If published, this will include your full peer review and any attached files.

Reviewer #1: **Yes: **Samuel Kofi Tchum

Reviewer #2: No

---

## [Author Response · Author response to Decision Letter 0]

27 Oct 2021

Response to Reviewers

A: Academic Editor

1. Acknowledgments/ Funding Section of your manuscript: The study was funded by Global Fund.

Response: The statement has been removed from the manuscript as instructed, and it has now been added through the online portal.

2. No authors have competing interests. Please complete your Competing Interests on the online submission form to state any Competing Interests.

Response: Done

3. PLOS requires an ORCID iD for the corresponding author in Editorial Manager on papers submitted after December 6th, 2016. Please ensure that you have an ORCID iD and that it is validated in Editorial Manager. 

Response: ORCID iD has been validated

B: Reviewer #1

My compliments to the authors, this is a well written manuscript about a properly designed and explained study. However, I have a few questions/remarks though:

1. Title: The title was not fluent. So I wanted to suggest “Malaria infection and anemia status in underfive South Tanzanian children where seasonal malaria chemoprevention is being implemented” instead of “Malaria infection and anemia status in underfive children in an area initiating seasonal malaria chemoprevention in Southern Tanzania”.

Response: The title suggested by the reviewer has been taken as it is, thank you.

2. Under introduction 

Justification and rationale for the study were not clearly stated. However, the introduction was more of information from literature. Please kindly review your introduction by elaborating on your justifications and rationales.

Response: Elaboration on justifications and rationales has been given, Lines 70-71, and 106-113.

3. Was the Ward randomization performed with a computer? Please elaborate. 

Response: A computer software, Research Randomizer was used to randomize wards. Line 149-150.

4. Under study population

I will suggest you rephrase it (P6, line 127) to mean willingness of the parents or caregivers to consent since these infants and your children are ethnically ineligible to consent for the study themselves.

Response: The sentence has been rephrased by adding the parents or caregivers, to indicate the willingness of parents or caregivers. Line 165. 

P6, line 128, please add the duration of stay in the study area.

Response: The duration has been added, the family should have lived in the study area for least 5 years. Line 168-69.

Under malariometric study 

o Please cite (P7, lines 148-149).

Response: The references number 29 and 30 have been added. 

o Please specify the number of readings used in the calculation for average parasite density (P8, lines 170- 171).

Response: The number of the readings used in the calculation for average parasite density has been explained, and now it reads, “the average parasite density of three readings was used in case of difference in parasite density of greater than 30%, whereas the average of the two positive readings was used for the case of positive versus negative results”. Lines 215-16.

o Please cite (P8, lines 174-175).

Response: Reference number 31 has been added. 

Under ethical consideration 

o Please cite (P8 line 194-195).

Response: Reference number 32 has been added. 

Under relationship between clinical presentation and malaria infection status

o The sentence is not clear (P12, lines 259-263).

Response: The sentence has been rephrased as suggested. Please clearly state whether it is positively or negatively associated (P13, line 266).

Response: Positively associated, and the word has been added in the sentence. Line 315.

o See Attachment for others minor errors.

Response: All the minor errors have been corrected.

C: Reviewer #2: 

Congratulations on the manuscript and the intense work carried out in the field.

Response: Thank you for the compliment.

---

## [Decision Letter · Decision Letter 1]

4 Nov 2021

PONE-D-21-21864R1Malaria infection and anemia status in underfive children from Southern Tanzania where seasonal malaria chemoprevention is being implemented.PLOS ONE

Dear Dr.  Mwaiswelo,

Thank you for resubmitting your manuscript to PLoS ONE. Although the data from this study has potential to be informative, relevant topics raised by the reviewer #2 during the peer review process remain to be addressed by the authors. At this time, we strongly suggest the authors to proper address all topics raised by the reviewers.  For your guidance, a copy of the reviewer’s comments was included below.  

We look forward to receiving your revised manuscript.

Kind regards,

Luzia Helena Carvalho, Ph.D.

Academic Editor

PLOS ONE

Journal Requirements:

Reviewers' comments:

Reviewer's Responses to Questions

**Comments to the Author**

1. If the authors have adequately addressed your comments raised in a previous round of review and you feel that this manuscript is now acceptable for publication, you may indicate that here to bypass the “Comments to the Author” section, enter your conflict of interest statement in the “Confidential to Editor” section, and submit your "Accept" recommendation.

Reviewer #1: All comments have been addressed

Reviewer #2: (No Response)

2. Is the manuscript technically sound, and do the data support the conclusions?

Reviewer #1: Yes

Reviewer #2: Yes

3. Has the statistical analysis been performed appropriately and rigorously? 

Reviewer #1: Yes

Reviewer #2: Yes

4. Have the authors made all data underlying the findings in their manuscript fully available?

Reviewer #1: Yes

Reviewer #2: Yes

5. Is the manuscript presented in an intelligible fashion and written in standard English?

Reviewer #1: Yes

Reviewer #2: Yes

6. Review Comments to the Author

Reviewer #1: The authors have adequately addressed my comments and suggestions raised in the previous round of review and I feel that this manuscript is now acceptable for publication. Also the authors made all data underlying the findings described in their manuscript fully available without restriction, with rare exception. Adherence to research and publication ethics maintained.

Reviewer #2: Dear author, I guess you might not have seen the final parts of my comments, so it was not approached by you in the answer. I am copying it right here again. It is very simple one.

1- It needs a better discussion about the presence of interfering or confounding variables in the results or even the complex causal relationship between socioeconomic data and malaria and anemia. When, socioeconomic conditions can be the cause of malaria and anemia, but also malaria and anemia are causes of low socioeconomic condition.

2- The manuscript needs language corrections, mainly in the introduction and methodology.(I believe, this one has already been settle).

7. PLOS authors have the option to publish the peer review history of their article (what does this mean?). If published, this will include your full peer review and any attached files.

Reviewer #1: **Yes: **Samuel Kofi Tchum

Reviewer #2: No

---

## [Author Response · Author response to Decision Letter 1]

11 Nov 2021

Response to Reviewers

Reviewer #2: 

1. Dear author, I guess you might not have seen the final parts of my comments, so it was not approached by you in the answer. I am copying it right here again. It is very simple one. 

It needs a better discussion about the presence of interfering or confounding variables in the results or even the complex causal relationship between socioeconomic data and malaria and anemia. When, socioeconomic conditions can be the cause of malaria and anemia, but also malaria and anemia are causes of low socioeconomic condition.

Response:

The discussion on causal relationship between socioeconomic data and malaria and anemia have been added on lines 330-333, 339-343, 385-389, 393-399.

2. The manuscript needs language corrections, mainly in the introduction and methodology.(I believe, this one has already been settle).

Response

Done.

---

## [Decision Letter · Decision Letter 2]

17 Nov 2021

Malaria infection and anemia status in under-five children from Southern Tanzania where seasonal malaria chemoprevention is being implemented.

PONE-D-21-21864R2

Dear Dr. Mwaiswelo,

We’re pleased to inform you that your manuscript has been judged scientifically suitable for publication and will be formally accepted for publication once it meets all outstanding technical requirements.

Kind regards,

Luzia Helena Carvalho, Ph.D.

Academic Editor

PLOS ONE

Additional Editor Comments (optional):

Reviewers' comments:

Reviewer's Responses to Questions

**Comments to the Author**

1. If the authors have adequately addressed your comments raised in a previous round of review and you feel that this manuscript is now acceptable for publication, you may indicate that here to bypass the “Comments to the Author” section, enter your conflict of interest statement in the “Confidential to Editor” section, and submit your "Accept" recommendation.

Reviewer #2: All comments have been addressed

2. Is the manuscript technically sound, and do the data support the conclusions?

Reviewer #2: Yes

3. Has the statistical analysis been performed appropriately and rigorously? 

Reviewer #2: Yes

4. Have the authors made all data underlying the findings in their manuscript fully available?

Reviewer #2: Yes

5. Is the manuscript presented in an intelligible fashion and written in standard English?

Reviewer #2: Yes

6. Review Comments to the Author

Reviewer #2: (No Response)

7. PLOS authors have the option to publish the peer review history of their article (what does this mean?). If published, this will include your full peer review and any attached files.

Reviewer #2: **Yes: **Dhelio Batista Pereira

---

## [Editor Report · Acceptance letter]

19 Nov 2021

PONE-D-21-21864R2 

Malaria infection and anemia status in under-five children from Southern Tanzania where seasonal malaria chemoprevention is being implemented. 

Dear Dr. Mwaiswelo:

I'm pleased to inform you that your manuscript has been deemed suitable for publication in PLOS ONE. Congratulations! Your manuscript is now with our production department. 

Kind regards, 

on behalf of

Dr. Luzia Helena Carvalho 

Academic Editor

PLOS ONE